# Bibliometric analysis of the global research trends and hotspots in chordoma from 2000 to 2020

Cuicui Chang[1☉], Kai Tang[1,2☉], Yifan Gao[1], Jingyao Dai[3]*, Chen Dai[4,5]*

**1** Central Laboratory of Medicine School, Xi'an Peihua University, Xi'an, China, **2** Department of Breast Surgery, The First Affiliated Hospital of Xi 'an Jiaotong University, Xi'an, China, **3** Department of Hepatobiliary Surgery, Air Force Medical Center, Fourth Military Medical University, Xi'an, China, **4** Orthopedics and Trauma Department, the 963rd (224th) Hospital of People's Liberation Army, 963rd Hospital of Joint Logistics Support Force of PLA, Jiamusi, Heilongjiang, China, **5** Department of Orthopedics, The Third Medical Center, General Hospital of the Chinese People's Liberation Army, Beijing, China

☉ These authors contributed equally to this work.

* daijingyao2006@163.com (JD); dcanb123@163.com (CD)

## Abstract

### Introduction

Chordoma is formed from embryonic residues or ectopic chordae and locally aggressive or malignant tumors. We visually analyzed the research tendency and hotspot of chordoma.

### Methods

The bibliometric analysis was conducted from the Web of Science Core Collection database over the past two decades. The term and strategies were as follows: "TS = (chordoma) OR TS = (chordoblastoma) OR TS = (chordocarcinoma) OR TS = (chordoepithelioma) OR TS = (chordosarcoma) OR TS = (notochordoma). AND Language: English. AND Reference Type: Article OR Review". A total of 2,118 references were retrieved and used to make a visual analysis by VOSviewer 1.6.15.

### Results

The chordoma was on a steady rise and chordoma but remained the focus of scholars and organizations over the last two decades. The Chinese institutions and scholars lacked cooperation with their counterparts in other countries. The citations of documents and co-citation analysis of cited references suggested that M.L. McMaster, B.P. Walcott, P. Bergh, and S. Stacchiotti were leading researchers in this field of chordoma and their papers had been widely accepted and inspired recent researches. Keywords associated with recent chemotherapy, PD-1-related immunotherapy, and SMARCB1/integrase interactor 1 (INI1) in chordoma were a shortage of research and there may be more research ideas in the future by scholars. The research of chordoma will continue to be the hotspot.

**Data Availability Statement:** The keywords of chordoma were indexed in the WOSCC (https://www.webofscience.com/wos/woscc/basic-search). The articles from 2000 to 2020 (December 31,

2020) were searched, the term "chordoma" was detected with MeSH (https://www.ncbi.nlm.nih.gov/mesh), and the term and strategy were as follows: "TS = (chordoma) OR TS = (chordoblastoma) OR TS = (chordocarcinoma) OR TS = (chordoepithelioma) OR TS = (chordosarcoma) OR TS = (notochordoma). AND Language: English. AND Reference Type: Article OR Review." A total of 2,118 references were retrieved and the results were analyzed visually.

**Funding:** This study was supported by the Shaanxi Oversea Scholars Scientific Research Foundation (NO. 2013SWZ01). The funder Jingyao Dai had role in study design, data analysis and decision to publish.

**Competing interests:** The authors have declared that no competing interests exist.

## Conclusions

Thus, explaining the molecular mechanism and potential role of transcriptional inhibition and immunologic responses to SMARCB1/INI1-negative poorly differentiated chordoma will be available for preclinical experiments and clinical trials and lead to new therapeutic opportunities for chordoma patients.

## Introduction

Chordoma is formed from embryonic residues or ectopic chordae and locally aggressive or malignant tumors that are common extradural tumors involving the ramus and sacrococcyx, with the overall incidence of 0.2–0.5/100,000 per year, accounting for about 0.15% of intracranial tumors, and about 40% of chordomas occurring in the sacral canal [1]. Five-year and ten-year survival rates were reported as low as 54.6 percent and 36.5 percent, respectively [1,2]. Although chordoma is slow growth, has few distant metastases (late metastases), and may delay diagnosis, its local destruction is strong, because of the high recurrence and metastasis rate after surgery (about 67% and 30%-40%, respectively). So, it is still a malignant tumor [1,3–6].

Considering that chordoma is insensitive to cytotoxic chemotherapy [1] and conventional radiotherapy (conventional radiotherapy usually only plays a palliative role) [4,7,8], radical surgery, such as the en bloc resection, is generally the treatment of choice in symptomatic cases [9]. Usually, the operation is difficult because [6,10]: (1) the anatomical location of chordoma is deep; (2) the surgical exposure is difficult; (3) the onset of chordoma is hidden and the course of the disease is long; and (4) when the patient comes to the clinic, the tumor has already extensively invaded the cranial base. Furthermore, complete resection is still the most critical factor in cancer metastasis, recurrence, and overall survival (OS) [1]. Subsequently, high-dose irradiation, particularly with protons and carbon ions, is a therapeutic alternative in cases of inoperable tumors [11]. Currently, targeted treatments have not been approved [12]. The lack of systematic therapeutic choices and deficient chordoma pathogenesis to direct the exploration of the innovative therapeutic strategies results in poor outcomes for patients with advanced disease [13].

It has been reported that various receptor tyrosine kinases are expressed in chordoma [14,15], including PDGFR-a/b, KIT, HER2, EGFR, VEGFR, and c-MET, etc., which activates downstream MAP kinase, PI3K/AKT/mTOR, and STAT3 signaling, etc. Recently, new insights into the molecular mechanisms of chordomas have led to the identification of novel potential treatments. The molecule-targeted therapies for chordomas include [15,16] (1) targeting PDGFR and KIT (imatinib and dasatinib); (2) targeting EGFR and HER2 (erlotinib, lapatinib, gefitinib, and cetuximab); (3) targeting VEGFR (sorafenib, pazopanib, and sunitinib); (4) targeting PI3K/AKT/mTOR pathway (temsirolimus and sirolimus). However, the systematic studies on the efficacy, safety, and molecular mechanisms of the molecule-targeted therapies for chordoma patients are still lacking [15]. Thus, we visually analyzed research tendency and hotspot of chordoma, by bibliometric analysis of the Web of Science Core Collection (WOSCC) database from 2000 to 2020. The bibliometric discusses the recent advancements of crosstalk between pathological features and molecular mechanisms and suggests the research tendency. It will point in a new direction for future study and clinical treatment of chordoma.

## Materials and methods

### Data collection

The keywords of chordoma were indexed in the WOSCC. The articles from 2000 to 2020 (December 31, 2020) were searched, the term "chordoma" was detected with MeSH (https://

www.ncbi.nlm.nih.gov/mesh), and the term and strategy were as follows: "TS = (chordoma) OR TS = (chordoblastoma) OR TS = (chordocarcinoma) OR TS = (chordoepithelioma) OR TS = (chordosarcoma) OR TS = (notochordoma). AND Language: English. AND Reference Type: Article OR Review." A total of 2,118 references were retrieved and the results were analyzed visually.

## Data analysis

Firstly, through the analysis and retrieval results in WOSCC, the general information of the literature, including the year of publication, country, organization, journal, author, and distribution of research field, were preliminarily analyzed. The number of published papers from different perspectives (year and country) was calculated by the online bibliometrics analysis website (http://bibliometric.com/), and the visual result showed the cooperative relationship between countries. Subsequently, VOSviewer software was performed to conduct bibliometric analysis and visualization analysis, including organizations, main authors, keywords, scientific research partnerships, cited analysis, and co-cited analysis. The standard tournament ranking method was adopted and the Method parameter in VOSviewer software was selected by Linlog/ modularization.

## Results

### Publication outputs

There were 2,118 items on chordoma in the WOSCC from 2000 to 2020 (December 31, 2020), included 1865 articles (88.05%) and 253 reviews (11.95%). The count of annual publications was shown in Fig 1, the publications appeared with 43 in 2020. Subsequently, the number of articles increased slowly year by year, and it was 197 in 2020 (Fig 1). Although the number of publications has been small but has been on a steady rise. The researches of chordoma were supported by 665 funding sources, and Table 1 showed the top 10 sources, such as National Institutes of Health NIH USA (frequency, 126), United States Department of Health Human Services (frequency, 216), National Natural Science Foundation of China NSFC (frequency, 90), NIH National Cancer Institute NCI (frequency, 90), and Ministry of Education Culture Sports Science and Technology Japan MEXT (frequency, 43), etc. The data suggest that chordoma remains the focus of scholars and organizations.

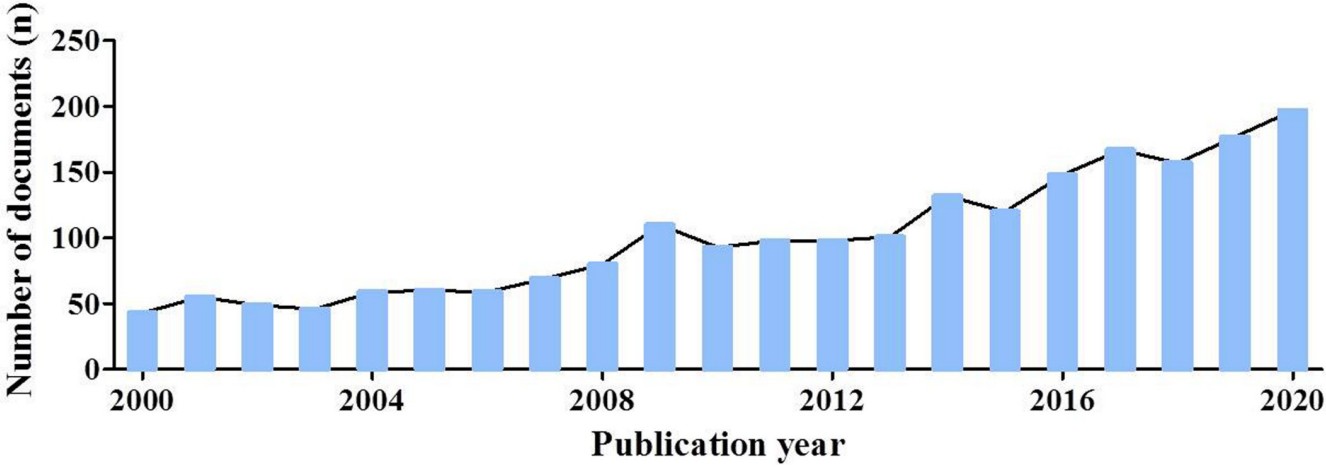

**Fig 1. Annual number of documents indexed in the WOSCC from 2000 to 2020 by the online bibliometric analysis.**

**Table 1. The top 10 funding sources.**

| Rank | Funding Source | Frequency |
|------|----------------|-----------|
| 1 | National Institutes of Health NIH USA | 126 |
| 2 | United States Department of Health Human Services | 126 |
| 3 | National Natural Science Foundation of China Nsfc | 90 |
| 4 | NIH National Cancer Institute NCI | 90 |
| 5 | Ministry of Education Culture Sports Science And Technology Japan Mext | 43 |
| 6 | Japan Society For The Promotion Of Science | 40 |
| 7 | Grants In Aid For Scientific Research Kakenhi | 34 |
| 8 | Chordoma Foundation | 23 |
| 9 | European Commission | 17 |
| 10 | Beijing Natural Science Foundation | 15 |

## Countries/regions and organization

A total of 60 countries/regions and 1,700 organizations had 2,118 articles on chordoma, which had been published in the last 20 years. USA (623), China (288), and Italy (148) were the top 3 countries of publications in chordoma in the WOSCC (Table 2). The USA had the most citation with 12,383, but the citation of publications in China were 2,652 that was weaker than that in Italy with 2,818 (Table 2). The cooperative relationship between countries was shown in Fig 2, the USA had more cooperation, but the total link strength and links of China were less than that in Italy, Germany, UK, and Canada (Fig 2). It indicates that there is a lack of cooperative researches between Chinese and other countries.

As shown in Table 3, the 10 major institutions were all from USA (80%) and China (20%). Massachusetts Gen Hosp (USA, 55), Capital Med Univ (China, 55), and Mayo Clin (USA, 48) were in the top 3 by the number of articles, but the top 3 by citations were Massachusetts Gen Hosp (1,402), Harvard Univ (1,364), and Mem Sloan Kettering Canc Ctr (984) (Table 3). Some articles were completed in collaboration with multiple institutions, as visualized (Fig 3). Each top organization showed extensive relationships with others (Fig 3), and 1,190 valid items (1,700 total items) were interlinked (Fig 3A). Some gray circles indicated that the institutions were isolated, such as Hosp Univ Canarias, Univ Montreal, and Taipei Med Univ, etc., which lacked partners (Fig 3A). The important partners of the top 3 organizations (minimum number of documents of an organization was 10, and 47 organizations met the thresholds) were analyzed in Fig 2B. Massachusetts Gen Hosp and Mayo Clin were each other's partners, and they were more than 75 partners (Fig 3B). However, the important partners of Capital Med Univ were almost the Chinese institutions, and Capital Med Univ had no relationship

**Table 2. Top 10 countries/regions on chordoma.**

| Rank | Country/Region | Documents | Citations | Total link strength | Links |
|------|----------------|-----------|-----------|---------------------|-------|
| 1 | USA | 623 | 12383 | 280 | 35 |
| 2 | China | 288 | 2652 | 69 | 13 |
| 3 | Italy | 148 | 2818 | 158 | 27 |
| 4 | Japan | 139 | 1913 | 61 | 19 |
| 5 | Germany | 106 | 2431 | 114 | 23 |
| 6 | UK | 101 | 1991 | 130 | 27 |
| 7 | Canada | 60 | 1102 | 99 | 18 |
| 8 | France | 59 | 1104 | 58 | 17 |
| 9 | Turkey | 58 | 446 | 7 | 4 |
| 10 | India | 58 | 423 | 15 | 9 |

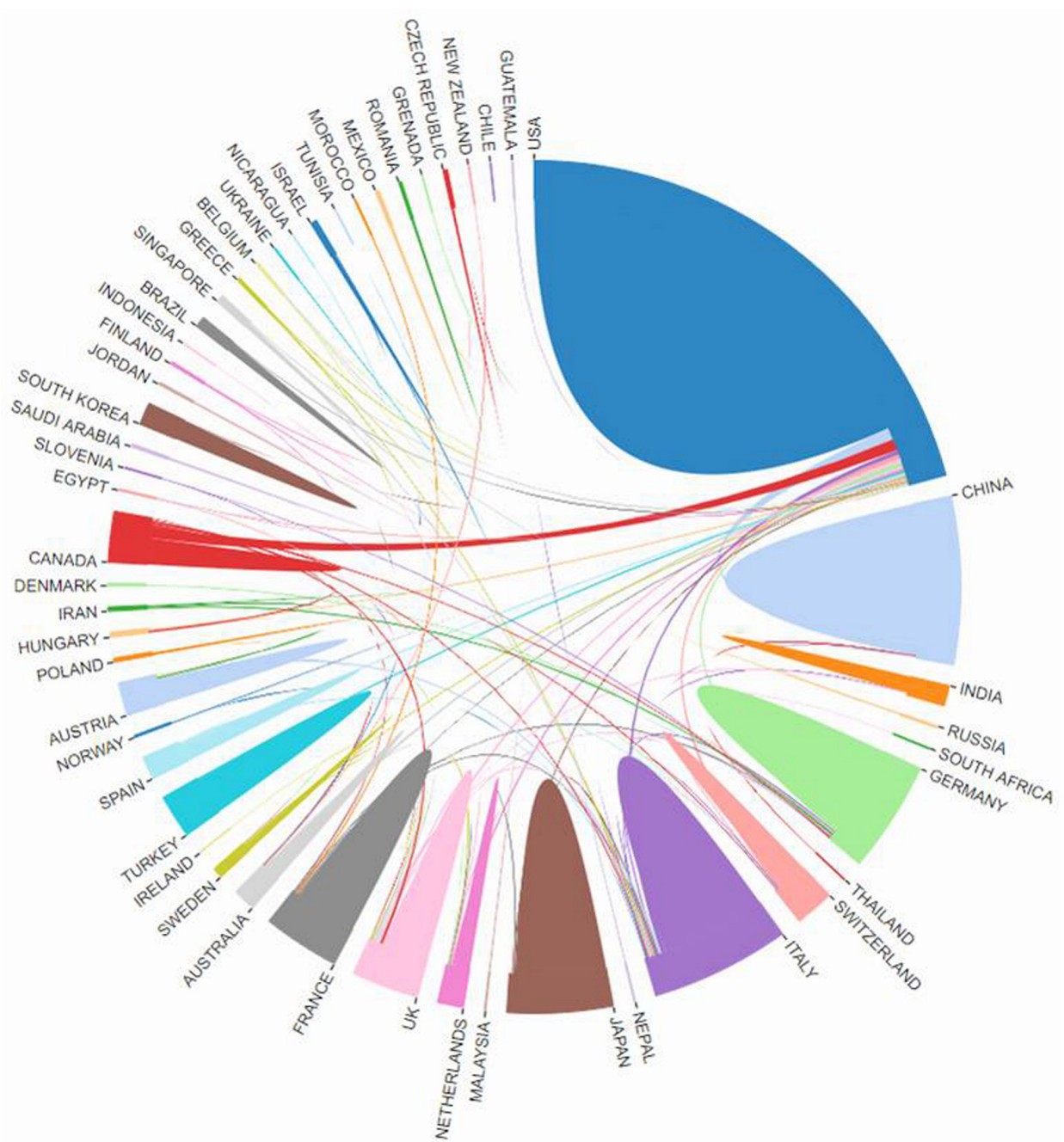

**Fig 2. The cooperative relationship between countries was generated by the online bibliometric analysis.**

with either Massachusetts Gen Hosp or Mayo Clin (Fig 3B). The data suggests the top institutions of chordoma researches lack more cooperation with each other and the Chinese institutions should improve cooperation with their counterparts in other countries.

## Journals analysis

In total, 1,638 journals published research documents related to chordoma from 2020 to 2020. In Table 4, the top 10 journals were shown that published about 25.12% of documents (532/

**Table 3. Top 10 the most productive organizations.**

| Rank | Organizations | Country | Documents | Citations | Total link strength |
|------|---------------|---------|-----------|-----------|---------------------|
| 1 | Massachusetts Gen Hosp | USA | 55 | 1402 | 147 |
| 2 | Capital Med Univ | China | 55 | 404 | 99 |
| 3 | Mayo Clin | USA | 48 | 700 | 118 |
| 4 | JohnsHopkins Univ | USA | 45 | 796 | 184 |
| 5 | Peking Univ | China | 41 | 545 | 43 |
| 6 | Univ Texas Md Anderson Canc Ctr | USA | 40 | 716 | 173 |
| 7 | Mem Sloan Kettering Canc Ctr | USA | 37 | 984 | 144 |
| 8 | Univ Calif San Francisco | USA | 33 | 836 | 91 |
| 9 | Harvard Med Sch | USA | 32 | 301 | 65 |
| 10 | Harvard Univ | USA | 28 | 1364 | 56 |

2,118). *World Neurosurgery* was the most dynamic journal of chordoma, followed by *Neurosurgery*, *Spine*, *Journal of Neurosurgery*, *European Spine Journal*, *International Journal of Radiation Oncology Biology Physics*, *Journal of Neurosurgery Spine*, *Acta Neurochirurgica*, *Journal of Clinical Neuroscience*, and *Journal of Neuro Oncology*. The impact factor (IF) of 10 journals was from 1.76 to 5.859, and International Journal of Radiation Oncology Biology Physics had the maximum IF of 5.859 (Q1), and *Journal of Clinical Neuroscience* had a minimum IF of 1.76 (Q3/Q4) (Table 4). But *Neurosurgery* was a good journal with 69 documents, 4.853 IF, and Q1 (Table 4). The selection of journals in chordoma was relatively wide, distributed in Q3/Q4 to Q1 (30%). According to the documents, IF and JCR partition, *Neurosurgery* may be the most popular journal in chordoma.

## Authors analysis

A total of 7,460 authors drafted the 2,118 documents in chordoma. In Table 5, Francis J Hornicek (Harvard Medical School) was the most active author in this field (with 46 documents and

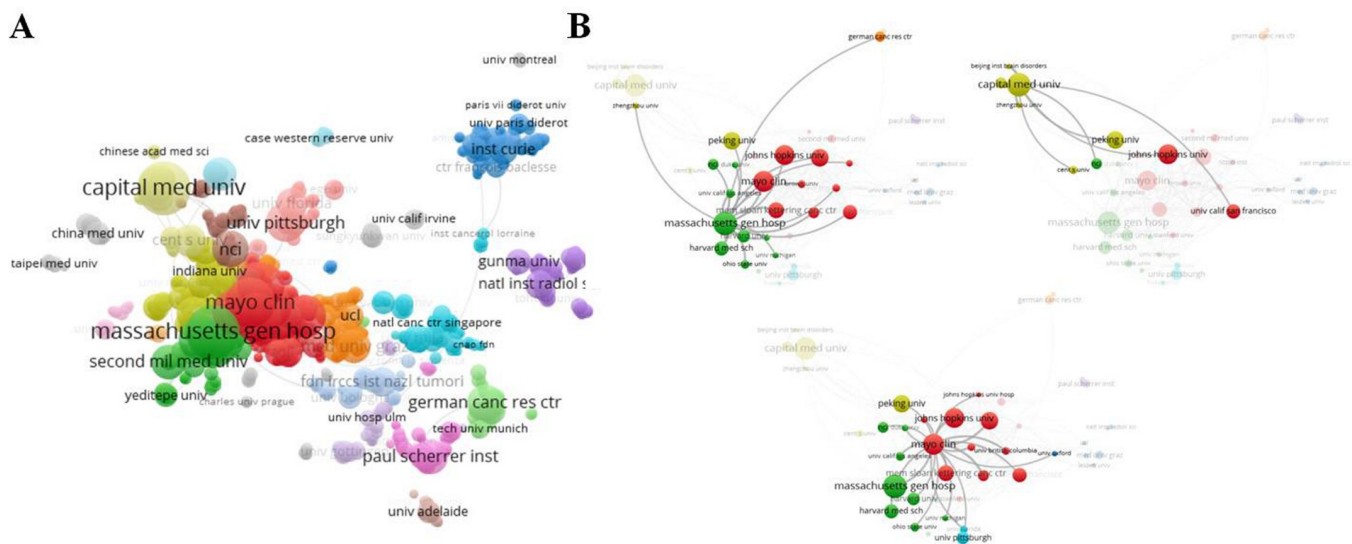

**Fig 3. Co-author analysis of organizations with network visualization.** (A) 1,190 valid items (1,700 total items) were interlinked. (B) The important partners of the top 3 organizations (minimum number of documents of an organization was 10, and 47 organizations met the thresholds). The thickness of lines indicates the strength of the relationship.

**Table 4. Top 10 journals with the largest number of publication.**

| Rank | Journals | Documents | 2019 Impact Factor | 2019 JCR Partition |
|---|---|---|---|---|
| 1 | World Neurosurgery | 84 | 1.829 | Q3 |
| 2 | Neurosurgery | 69 | 4.853 | Q1 |
| 3 | Spine | 64 | 2.646 | Q2 |
| 4 | Journal of Neurosurgery | 60 | 3.968 | Q1 |
| 5 | European Spine Journal | 52 | 2.458 | Q2/Q3 |
| 6 | International Journal of Radiation Oncology Biology Physics | 51 | 5.859 | Q1 |
| 7 | Journal of Neurosurgery Spine | 50 | 3.011 | Q1/Q2 |
| 8 | Acta Neurochirurgica | 38 | 1.817 | Q3 |
| 9 | Journal of Clinical Neuroscience | 33 | 1.76 | Q3/Q4 |
| 10 | Journal of Neuro Oncology | 31 | 3.267 | Q2/Q3 |

1,023 citations), Ziya L Gokaslan (Brown University), Joseph H Schwab (Harvard Medical School), Zhen Wu (Capital Medical University), and Liang Wang (Capital Medical University) followed. Ke Wnag had the minimum citations 170 in this ranking, but lower-ranking Jürgen Debus and Jean-Paul Wolinsky, professional and active authors in this field, also had high citations 668 and 446, respectively (Table 5). The co-authorship map of authors was generated by VOSviewer (Fig 3), and 2,918 valid items (7,460 total items) were connected (Fig 3A). Some researchers were also scattered independently with other activated scholars and Francis J Hornicek was the center but failed to reach out to all the scattered cliques (Fig 4A). In Fig 4B, 203 organizations (more than 5 documents) met the thresholds. Francis J Hornicek, Ziya L Gokaslan, and Joseph H Schwab had some cooperation, but Zhen Wu and Liang Wang were limited to their small teamwork (purple circle) (Fig 4A). The data suggests that the activated authors in chordoma still lack collaboration with other scholars around the world.

## Citation analysis

There were 25,969 cited references, the front-ranking were M.L. McMaster (Cancer Causes Control, 2001), B.P. Walcott (Lancet Oncol, 2012), P. Bergh (Cancer, 2000), R. Chugh (Oncologist, 2007), S. Vujovic (J Pathol, 2006), B. Fuchs (J Bone Joint Surg Am, 2005), S. Boriani (Spine, 2006), J.E. York (Neurosurgery, 1999), S. Stacchiotti (Lancet Oncol, 2015), and C. Catton (Radiother Oncol, 1996,). "Chordoma: incidence and survival patterns in the United States, 1973–1995" was the highest cited reference in the field of chordoma, with 373 citations, and "Soft tissue sarcoma of the extremity. Limb salvage after the failure of combined

**Table 5. Top 10 active authors with most documents.**

| Authors | Organizations | Documents | Citations | h-index |
|---|---|---|---|---|
| Francis J. Hornicek | Harvard Medical School | 46 | 1023 | 57 |
| Ziya L. Gokaslan | Brown University | 40 | 883 | 64 |
| Joseph H. Schwab | Harvard Medical School | 38 | 794 | 49 |
| Zhen Wu | Capital Medical University | 27 | 266 | 19 |
| Liang Wang | Capital Medical University | 25 | 244 | 13 |
| Daniel M. Sciubba | Johns Hopkins University | 24 | 373 | 43 |
| Stefano Boriani | IRCCS Istituto Ortopedico Galeazzi | 23 | 379 | 37 |
| Ke Wnag | Capital Medical University | 22 | 170 | 14 |
| Jürgen Debus | University of Heidelberg | 21 | 668 | 75 |
| Jean P. Wolinsky | Northwestern University | 20 | 446 | 37 |

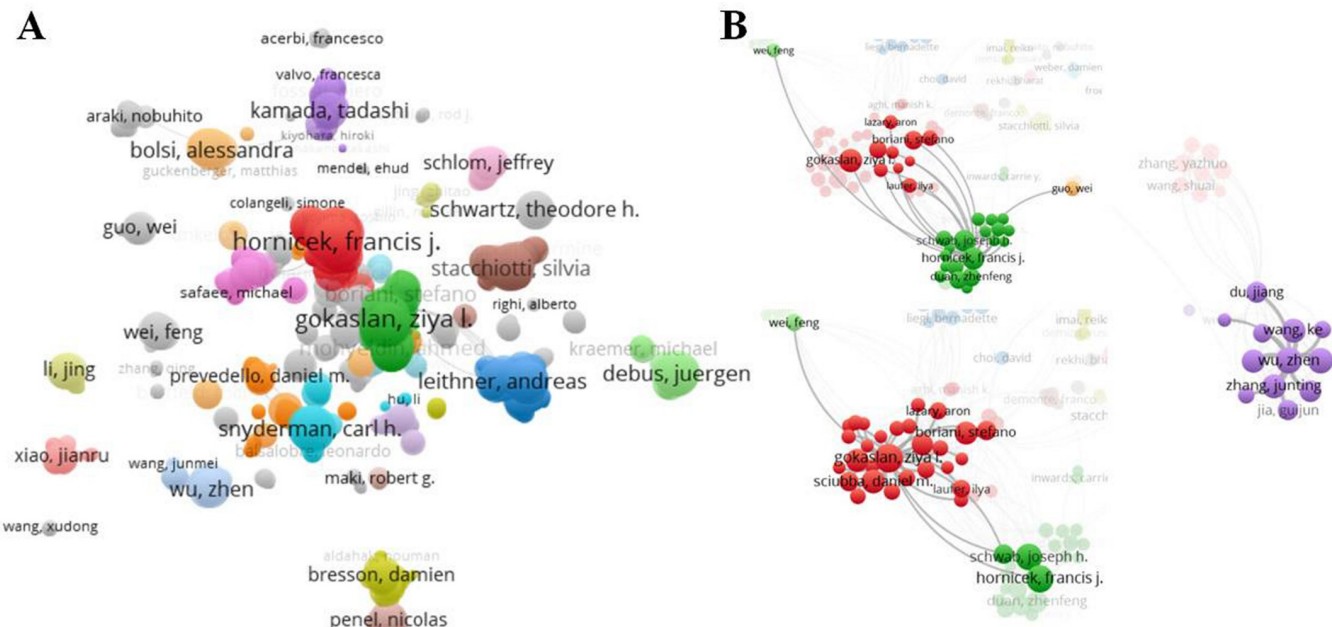

**Fig 4. Co-occurrence analysis of authors.** (A) 2,918 valid items (7,460 total items) were connected. (B) The cooperation of the top 3 authors was presented (the minimum number of documents of an organization was 10, and 47 organizations met the thresholds).

conservative therapy" was the minimum with 130 citations (Table 6). Generally, most of the cited literature is based on clinical studies of chordoma. As shown in Table 7, the top 3 citations of documents were "Chordoma: incidence and survival patterns in the United States, 1973–1995" by M.L. McMaster, "Chordoma: current concepts, management, and future directions" by B.P. Walcott, and "Prognostic factors in chordoma of the sacrum and mobile" by P. Bergh, which corresponded to the top 3 highest cited references in Table 6. Two articles were from Lancet Oncology by B.P. Walcott (2012, 325 citations) and S. Stacchiotti (2015, 180 citations), and two articles were from *International Journal of Radiation Oncology Biology Physics*

**Table 6. Top 10 Co-citation of cited reference on chordoma.**

| Rank | Title | First author | Source | Publication year | Total citations |
|---|---|---|---|---|---|
| 1 | Chordoma: incidence and survival patterns in the United States, 1973–1995 | M. L. McMaster | Cancer Causes Control | 2001 | 373 |
| 2 | Chordoma: current concepts, management, and future directions | B. P. Walcott | Lancet Oncol | 2012 | 256 |
| 3 | Prognostic factors in chordoma of the sacrum and mobile spine: a study of 39 patients | P. Bergh | Cancer | 2000 | 220 |
| 4 | Chordoma: the nonsarcoma primary bone tumor | R. Chugh | Oncologist | 2007 | 187 |
| 5 | Brachyury, a crucial regulator of notochordal development, is a novel biomarker for chordomas | S. Vujovic | J Pathol | 2006 | 175 |
| 6 | Operative management of sacral chordoma | B. Fuchs | J Bone Joint Surg Am | 2005 | 170 |
| 7 | Chordoma of the mobile spine: fifty years of experience | S. Boriani | Spine | 2006 | 167 |
| 8 | Sacral chordoma: 40-year experience at a major cancer center | J. E. York | Neurosurgery | 1999 | 144 |
| 9 | Building a global consensus approach to chordoma: a position paper from the medical and patient community | S. Stacchiotti | Lancet Oncol | 2015 | 131 |
| 10 | Soft tissue sarcoma of the extremity. Limb salvage after failure of combined conservative therapy | C. Catton | Radiother Oncol | 1996 | 130 |

**Table 7. Top 10 citation analysis of documents on chordoma.**

| Rank | Title | First author | Source | Publication year | Total citations |
|---|---|---|---|---|---|
| 1 | Chordoma: incidence and survival patterns in the United States, 1973–1995 | M.L. McMaster | Cancer Causes Control | 2001 | 544 |
| 2 | Prognostic factors in chordoma of the sacrum and mobile spine | P. Bergh | Cancer | 2000 | 368 |
| 3 | Chordoma: current concepts, management, and future directions | B.P. Walcott | Lancet Oncol | 2012 | 325 |
| 4 | EWSR1-POU5F1 fusion in soft tissue myoepithelial tumors. A molecular analysis of sixty-six cases, including soft tissue, bone, and visceral lesions, showing common involvement of the EWSR1 gene | C.R. Antonescu | Genes Chromosomes Cancer | 2010 | 277 |
| 5 | Image-guided Hypo-fractionated Stereotactic Radiosurgery to Spinal Lesions | S.I. Ryu | Neurosurgery | 2001 | 232 |
| 6 | Building a global consensus approach to chordoma: a position paper from the medical and patient community | S. Stacchiotti | Lancet Oncol | 2015 | 180 |
| 7 | Intensity modulated proton therapy: A clinical example | A.J. Lomax | Med Phys | 2001 | 169 |
| 8 | Stereotactic fractionated radiotherapy for chordomas and chondrosarcomas of the skull base | J. Debus | Int J Radiat Oncol Biol Phys | 2000 | 162 |
| 9 | Effectiveness and Safety of Spot Scanning Proton Radiation Therapy for Chordomas and Chondrosarcomas of the Skull Base: First Long-Term Report | C. Ares | Int J Radiat Oncol Biol Phys | 2009 | 158 |
| 10 | Phase II Study of Imatinib in Advanced Chordoma | S. Stacchiotti | J Clin Oncol | 2012 | 148 |

by J. Debus (2000, 162 citations) and Carmen Ares (2009, 158 citations) (Table 7). S. Stacchiotti had two documents in this ranking (Lancet Oncology, 2015, 180 citations; Journal of Clinical Oncology, 2012, 148 citations) (Table 7). So, M.L. McMaster, B.P. Walcott, P. Bergh, and S. Stacchiotti were leading researchers in this field of chordoma and their papers deserved wide attention. 1,618 documents were cited among 2,118 articles, and 1,438 valid items were connected and the overlay or network visualization was conducted (Fig 5). The overlay visualization showed that some articles have recently been noted and cited (Fig 5A). Fig 5B showed

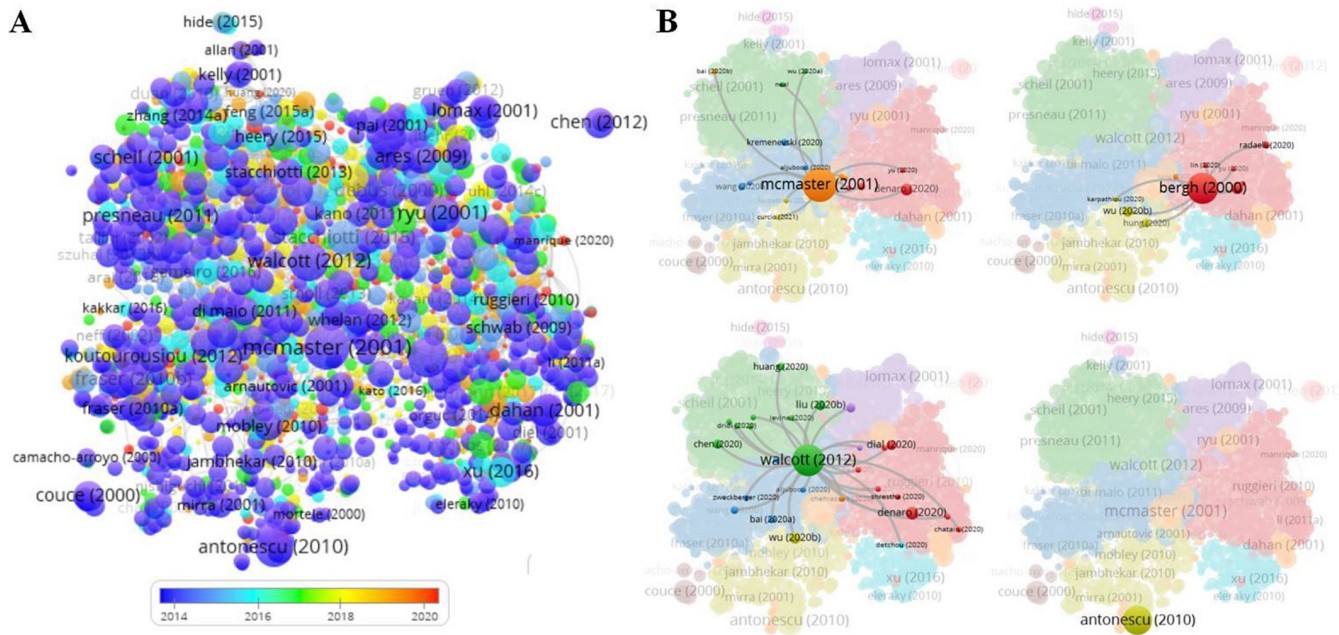

**Fig 5. Citation analysis of documents.** 1,618 documents were cited among 2,118 articles, and 1,438 valid items were connected. (A) The overlay visualization of citation map of documents was generated by Linlog/modularity in VOSviewer, the Weight was citations, and Scores were the average published year. (B) The network visualization was conducted, and the crosstalk of citation documents was presented.

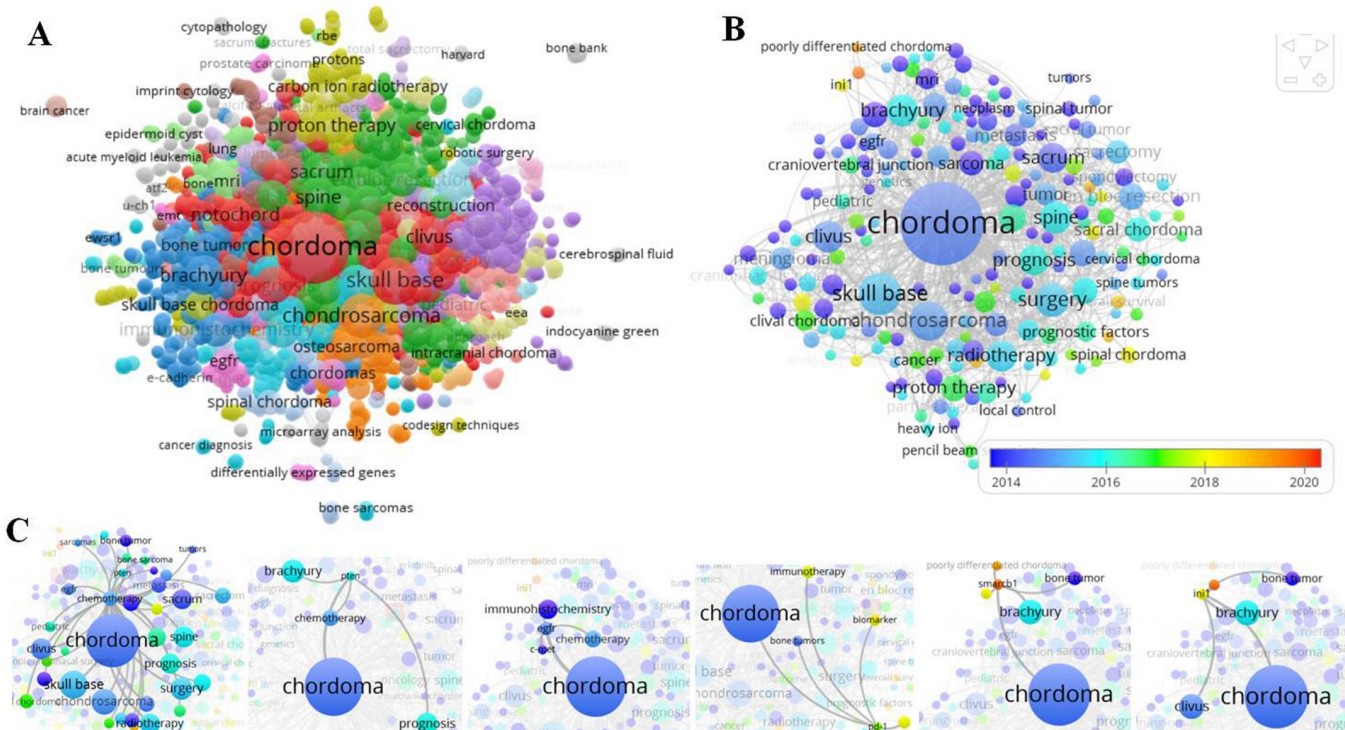

**Fig 6. Co-occurrence analysis of author keywords.** (A) The network visualization of total author keywords (2,656 items) was conducted. (B) The overlay visualization of co-occurrence analysis of author keywords was presented in VOSviewer, the Weight was an occurrence, and Scores were the average published year. (C) The crosstalk of author keywords was presented.

the relationship between the high citation articles by M.L. McMaster (2001), B.P. Walcott (2000), P. Bergh (2012), and C.R. Antonescu (2010) with other articles. "EWSR1-POU5F1 fusion in soft tissue myoepithelial tumors. A molecular analysis of sixty-six cases, including soft tissue, bone, and visceral lesions, showing" by C.R. Antonescu was distant from recent highly cited articles, but the other 3 articles were closely related to the recent highly cited articles (Fig 5B). All those data suggest some articles of chordoma with high citations have been widely accepted and have inspired recent researches, and recent studies should improve scale and breakthrough.

## Keywords analysis

There were 2,781 author keywords involved in 2,118 articles, and 2,656 valid items were connected (Fig 6A). The co-occurrence relations of keywords and reflected the hot and frontier topics in chordoma. Fig 6A presented the biggest keywords subnetwork and its cluster map. The "chordoma", "skull base", and "chondrosarcoma" were the top 3 occurrences of keywords, and they related to the keywords of "surgery", "clivus", "brachyury (TBXT)", "spine", "radiotherapy", "recurrence", "immunohistochemistry", etc. With a minimum of 5 number occurrences of a keyword, 102 keywords met the threshold with overlay or network visualization (Fig 6B and 6C). In the overlay visualization, keywords mainly appeared in the decade between 2014 and 2018, but recent studies reported less, such as SMARCB1 and integrase interactor 1 (INI1) (Fig 6B). Fig 6C showed that chemotherapy, PTEN, EGFR, C-Mte, PD-1, immunotherapy, SMARCB1, and INI1 were related to chordoma. SMARCB1/INI1 was also involved in TBXT and bone tumor (Fig 6C). The data show recent PD-1-related immunotherapy and

SMARCB1/INI1 in chordoma were a shortage in research and there may be more research ideas in the future by scholars.

## Discussion

Over the last two decades, the researches on chordoma were on a steady rise and chordoma remained the focus of scholars and organizations. Although many countries, organizations, and scholars were involved in chordoma, the activated organizations and authors, especially the top teams in this field, still lacked collaboration with others around the world. The citations of documents and co-citation analysis of cited references suggested that M.L. McMaster, B.P. Walcott, P. Bergh, and S. Stacchiotti were leading researchers in this field of chordoma and their papers had been widely accepted and inspired recent researches, but further studies should improve scale and breakthrough. Keywords associated with recent chemotherapy, PD-1-related immunotherapy, and SMARCB1/INI1 in chordoma were a shortage of research and there may be more research ideas in the future by scholars. Bibliometric is based on literature system, such as the WOSCC database, is different from a literature review and often applied to analyze the hot spots, research progress and other related issues, as well as to formulate future research directions and strategies. Through visual analysis, the cooperative relationship between organizations, countries and authors and network of research directions, which allows identification of key opinion leaders and future research directions to help in fostering optimised future collaborative networks. Based on the analysis, we recommend that explaining the molecular mechanism and potential treatments will be available for preclinical experiments and clinical trials and lead to new therapeutic opportunities for chordoma patients.

Bibliometric analysis showed that chemotherapy, PTEN, EGFR, and TBXT were related to chordoma. Targeting PTEN, EGFR, C-Mte, and PD-1 can provide potential strategies for the treatment of chordoma. report that the expression of PTEN is higher in adjacent normal tissues than that in sacral chordomas, and PTEN-negative expression and mTOR-positive expression are related to the invasion of tumor [17]. Loss of PTEN gene heterozygosity in chordoma subpopulations is associated with aggressive behavior and increased Ki-67 proliferation index in vitro [18], and the combined suppression of PDGFR and HDAC subsequently decreases the proliferation and invasion of PTEN deficient chordoma cells. Clinical data further demonstrate that the patients with higher PTEN expression show distinctly longer OS and progression-free survival (PFS) than those with lower levels [19], and PTEN expression may serve as a prognostic and predictive biomarker for chordomas. It has been reported that FGF2 and TGF-a involve in chordoma recurrence [20] and FGFR/MEK/ERK/TBXT pathway coordinately regulates chordoma cell growth and survival [21]. Morimoto et al. have demonstrated that the expression of VEGFR1 and VEGFR2 on tumor cells and immunosuppressive tumor-microenvironment were positively correlated with tumor growth and recurrent in patients with chordomas [22]. EGFR inhibitors were regarded as a potential therapy for chordoma [23]. Patients with higher TBXT expression have significantly shorter progression-free survival than those with lower expression [24], and TBXT inhibits Paclitaxel-induced apoptosis in primary chordoma cell lines via carbonic anhydrase IX [25]. Pharmacologic inhibition of H3K27-demethylases promotes human chordoma cell death via inducing epigenetic silencing of oncogenic TBXT [26]. Afatinib has been verified as a potential treatment for chordomas by targeting EGFR and TBXT [27]. Sharifnia et al. demonstrate that inhibitors of CDK suppress TBXT levels and chordoma tumor proliferation is decreased by THZ1 treatment in vivo [13]. However, it is challenging for transcription factors, including TBXT, to become candidate drugs for direct targets [13]. Transcriptional EGFR and CDK suppression can provide therapeutic opportunities for preferential down-regulation of TBXT [13]. Thus, the recent entry of

a new generation of transcriptional inhibitors for clinical application may suggest a much-needed treatment option for this refractoriness disease in patients with chordoma.

Positive PD-L1 expression on tumor cells is associated with advanced stages [28], and the density of PD-1+ tumor-infiltrating lymphocytes is associated with the invasion of surrounding muscles [29,30]. Comparative analysis reveals higher levels of soluble forms of PD-L1 in the blood serum of chordoma patients in comparison with healthy persons, but the soluble form of PD-1 is at the same level in both chordoma patients and healthy persons [31]. So, immunotherapy of PD-1/ L1 may be a potential therapy in advanced patients of chordoma. In a case report, a patient with chordoma receives pembrolizumab and achieves a PFS of 9.3 months by targeting PD-1 receptors of lymphocytes [32], but the immunotherapy benefit may be related to the A1209fs mutation of the PBRM1 gene [32]. Thus, the effectiveness and application of immunotherapy should be further explored and clarified in the patient response mechanism of chordoma.

As one of the core subunits of the ATP-dependent SWI/SNF chromatin remodeling complex, SMARCB1/INI-1 is regarded as a suppressor of tumors [33]. Recent studies explore the role of SMARCB1/INI-1 in chordoma, and the loss of SMARCB1/INI1 protein in poorly differentiated chordoma (PDC) associates not with point mutations but with SMARCB1/INI1 gene deletions instead [34]. The PDC, a tumor that often occurs in children, prefers the base of the skull and appears nuclear TBXT positivity along with INI-1 loss [34,35]. Case analysis findings suggest that a SMARCB1 deletion is an early event in the rare traditional chordoma that may transform into PDC through supernumerary genome variation [36]. SMARCB1/INI1-negative PDC is a unique subset of chordomas with clinical, histopathological, and molecular specificity, rapid progression, and poor prognosis, and should not be confused with traditional chordomas [37,38]. SMARCB1 expression is completely absent in both PDC and atypical teratoid/rhabdoid tumor (AT/RT), and TBXT and cytokeratin immunoexpressions of PDC are diffuse and strong, and combining SMARCB1 with TBXT staining helps to distinguish PDC from AT/RT [39]. In the miRNA-SMARCB1/INI1 regulatory network, Malgulwar et al. show miR-193a-5p and miR-671-5p mediate the acquired SMARCB1/INI1 deletion and downregulate the TGF-β pathway in the pediatric chordoma [40]. The results of a clinical trial are exciting that EZH2 binding tazemetostat provides an anti-tumor immunoreaction via inducing multiple T cell populations to infiltrate tumors and demonstrates the potential effect on the epigenetic regulation of immunoreaction in SMARCB1/INI1-negative PDC and this patient also has a benefit in OS [41]. The data provide new references for revealing the regulatory mechanisms and treatment strategies in SMARCB1/INI1-negative PDC, such as epigenetic regulation.

Some limitations were addressed in this analysis. Firstly, the data was collected on December 31, 2020, but WOSCC database continued to be updated during 2021, and this part was omitted in this study. Besides, the search terms "TS = (chordoma) OR TS = (chordoblastoma) OR TS = (chordocarcinoma) OR TS = (chordoepithelioma) OR TS = (chordosarcoma) OR TS = (notochordoma). AND Language: English. AND Reference Type: Article OR Review." were selected to define the topic of the studies in WOSCC database, not all documents were completely obtained. Thirdly, as the data was limited to WOSCC indexed journal, some data lacked in the WOSCC database. But we believe this analysis can still be employed to describe general trends and hotspots of chordoma.

## Conclusion

In summary, the work discussed chordoma research conditions by bibliometric analysis. The chordoma, as a branch of bone tumor, was in a steady rise and chordoma but remained the

focus of scholars and organizations over the last two decades. The Chinese institutions and scholars should improve cooperation with their counterparts in other countries. Based on the pre-efforts and exploration of PD-1-related immunotherapy and SMARCB1/INI1 in chordoma by a wide range of scholars, further studies should improve scale and breakthrough. Mechanically, explaining the molecular mechanism and potential role of transcriptional inhibition and immunologic responses to SMARCB1/INI1-negative PDC will be available for preclinical experiments and clinical trials and lead to new therapeutic opportunities for chordoma patients.

## Author Contributions

**Conceptualization:** Cuicui Chang, Jingyao Dai, Chen Dai.

**Data curation:** Kai Tang, Yifan Gao.

**Formal analysis:** Yifan Gao, Jingyao Dai.

**Funding acquisition:** Jingyao Dai.

**Methodology:** Cuicui Chang, Kai Tang, Yifan Gao.

**Supervision:** Chen Dai.

**Writing – original draft:** Cuicui Chang.

**Writing – review & editing:** Chen Dai.

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
