## [Decision Letter · Decision Letter 0]

15 Aug 2022

PONE-D-21-36906Bibliometric analysis of the global research trends and hotspots in chordoma from 2000 to 2020PLOS ONE

Dear Dr. Dai,

Thank you for submitting your manuscript to PLOS ONE. After careful consideration, we feel that it has merit but does not fully meet PLOS ONE’s publication criteria as it currently stands. Therefore, we invite you to submit a revised version of the manuscript that addresses the points raised during the review process.

- Please revise the manuscript per the reviewers' comments.

- Please follow the PLoS One guideline to prepare the manuscript.

- The manuscript requires English copy editing. Please attach the certificate.

We look forward to receiving your revised manuscript.

Kind regards,

Farzad Taghizadeh-Hesary

Academic Editor

PLOS ONE

Journal Requirements:

2. Thank you for stating the following financial disclosure: "This study was supported by the Shaanxi Oversea Scholars Scientific Research Foundation (NO. 2013SWZ01)."

Please state what role the funders took in the study.  If the funders had no role, please state: "The funders had no role in study design, data collection and analysis, decision to publish, or preparation of the manuscript.

4. Please upload a new copy of Figures 2-6 as the detail is not clear. Please follow the link for more information: https://blogs.plos.org/plos/2019/06/looking-good-tips-for-creating-your-plos-figures-graphics/" https://blogs.plos.org/plos/2019/06/looking-good-tips-for-creating-your-plos-figures-graphics/

Reviewers' comments:

Reviewer's Responses to Questions

**Comments to the Author**

1. Is the manuscript technically sound, and do the data support the conclusions?

Reviewer #1: Yes

Reviewer #2: Yes

2. Has the statistical analysis been performed appropriately and rigorously? 

Reviewer #1: Yes

Reviewer #2: I Don't Know

3. Have the authors made all data underlying the findings in their manuscript fully available?

Reviewer #1: Yes

Reviewer #2: Yes

4. Is the manuscript presented in an intelligible fashion and written in standard English?

Reviewer #1: Yes

Reviewer #2: Yes

5. Review Comments to the Author

Reviewer #1: This is an interesting study to examine the publish pattern in chordoma in the last two decade utilizing “bibliometric analysis” , which provides some additional and interesting information compared to what we used to see in systemic review or review article.

However I do have some concerns and recommendation for authors.

1. I did a pubmed search using same time period from Jan 1 2000 to Dec 31 2020 but just chordoma as a keyword and I found 2894 articles and 454 reviews. The authors used a wide range of keywords and many of which may not be chordoma diagnosis but only 1865 articles and 253 reviews retrieved. Authors should explain the difference of WOSCC versus other public available most used scientific databases. In addition, authors should clarify why other key words were also selected and those articles did not focus on chordoma should be excluded.

2. Although it is very interesting to see countries, regions, journals, authors and citations of publications in the global fashion in the existing literature, I think that authors could further highlight the most important advancement/discovery that has been made in the last two decades in four categories (pathology diagnosis, surgery, radiation, and systemic treatment). Authors did a good job on last one but should do similar discussion on other arears as well. Also readers would be very interested to know the consistency or discrepancy from different publications especially from different countries/regions.

3. From the methodology perspective, authors should emphasize the unique advantages or differences of bibliometric analysis versus systemic review as this analysis is not as common and maybe unfamiliar to most readers. More importantly this could address what unique contribution of this particular article has generated in this field by addressing this issue.

4. It is notable that most journals publishing most chordoma paper in the last two decades are relatively low impact. I think it is important for authors to discuss this important finding which may support the future research for rare cancer.

5. Although this is well-written manuscript and authors address points clearly and concisely but I strongly recommend authors to use professional English writer (or use Grammars software) to polish up English throughout. One example “ Although chordoma is slow growth, few distant metastases (late metastases), and may delay diagnosis, its local destruction…. “ this sentence does not make sense. Another example : Many places authors use “then” which could be replace by “subsequent” as more professional writing.

Reviewer #2: The authors of this article have attempted to conduct a bibliometric analysis of Chordoma and the current research regarding the same. They also have identified the lacunae in the chinese publications and their lack of of co-operative research. This knowledge is probably relevant to the readers in china and does give some idea to the asian and south asian neurosurgical research communities.

6. PLOS authors have the option to publish the peer review history of their article (what does this mean?). If published, this will include your full peer review and any attached files.

Reviewer #1: **Yes: **Xiaolan Feng

Reviewer #2: No

---

## [Author Response · Author response to Decision Letter 0]

16 Aug 2022

Reviewer #1: This is an interesting study to examine the publish pattern in chordoma in the last two decade utilizing “bibliometric analysis” , which provides some additional and interesting information compared to what we used to see in systemic review or review article.

However I do have some concerns and recommendation for authors.

1. I did a pubmed search using same time period from Jan 1 2000 to Dec 31 2020 but just chordoma as a keyword and I found 2894 articles and 454 reviews. The authors used a wide range of keywords and many of which may not be chordoma diagnosis but only 1865 articles and 253 reviews retrieved. Authors should explain the difference of WOSCC versus other public available most used scientific databases. In addition, authors should clarify why other key words were also selected and those articles did not focus on chordoma should be excluded.

Response: Thank you for your professional suggestion. 

Bibliometric is based on literature system, such as the WOS Core Collection database [1-6], using mathematics and statistics methods to study the distribution structure and quantitative relationship of literature , which is similar to statistical research in methodology and often applied to analyze the hot spots, research progress and other related issues, as well as to formulate future research directions and strategies. We just only retrieved most representative articles for visual and further to study development status, development trends of related articles and predict the research hotspots in future. We do this work is to provide a guidance for ourselves also others who engaged in chordoma related field. It’s like a sample survey in statistics, samples drawn from population can represent the characteristics of population in a certain extent. These features can help us recognize population in a certain extent. One of the calculated methods used in bibliometric is statistics. Hence in principle, representative articles we drawn is similar with samples drawn from population, they can explain the development status and development trends in a certain degree. Of course, there are still PubMed, Wiley, MEDLINE, and other databases to refer to. This is also a limitation of our work, which we have stated in the manuscript.

In this work, the bibliometric analysis was conducted from the Web of Science Core Collection database from 2000 to 2020 (December 31, 2020). The term “chordoma” was detected with MeSH (https://www.ncbi.nlm.nih.gov/mesh), which included “chordoma”, “chordoblastoma”, “chordocarcinoma”, “chordoepithelioma”, “chordosarcoma”, and “notochordoma”. The term and strategy were as follows: “TS=(chordoma) OR TS=(chordoblastoma) OR TS=(chordocarcinoma) OR TS=(chordoepithelioma) OR TS=(chordosarcoma) OR TS=(notochordoma). AND Language: English. AND Reference Type: Article OR Review”. 

Web of Science Core Collection has secondary retrieval function and we can refine or exclude articles according to our own needs. By the Clarivate Analytics WOS Core Collection, the subset could not be excluded, and we selected articles and reviews published in English from 2000 to 2020. So the Commentaries, Patents, Abstracts, and Conferences were excluded. A total of 2,118 references were retrieved, and then were used to make a visual analysis by VOSviewer 1.6.15.

1. Reddy VP, Singh R, McLelland MD, Barpujari A, Catapano JS, Srinivasan VM, Lawton MT. Bibliometric Analysis of the Extracranial-Intracranial Bypass Literature. World Neurosurg. 2022;161:198-205.e5.

2. Berta A, Miguel Ángel C, Clara GS, Rubén H. A bibliometric analysis of 10 years of research on symptom networks in psychopathology and mental health. Psychiatry Res. 2022;308:114380.

3. Wei Q, Shen J, Wang D, Han X, Shi J, Zhao L, Teng Y. A bibliometric analysis of researches on flap endonuclease 1 from 2005 to 2019. BMC Cancer. 2021;21(1):374.

4. Vuillemin N, Pape HC, Rommens PM, Lippuner K, Siebenrock KA, Keel MJ, Bastian JD. A Bibliometric Analysis of Fragility Fractures: Top 50. Medicina (Kaunas). 2021;57(6):639.

5. Yang W, Liu Y, Zeng T, Wang Y, Hao X, Yang W, Wang H. Research focus and thematic trends in magnet hospital research: A bibliometric analysis of the global publications. J Adv Nurs. 2021;77(4):2012-2025.

6. Fares J, Chung KSK, Abbasi A. Stakeholder theory and management: Understanding longitudinal collaboration networks. PLoS One. 2021 Oct 14;16(10):e0255658.

2. Although it is very interesting to see countries, regions, journals, authors and citations of publications in the global fashion in the existing literature, I think that authors could further highlight the most important advancement/discovery that has been made in the last two decades in four categories (pathology diagnosis, surgery, radiation, and systemic treatment). Authors did a good job on last one but should do similar discussion on other arears as well. Also readers would be very interested to know the consistency or discrepancy from different publications especially from different countries/regions.

Response: Thank you for your professional suggestion. 

Bibliometric is based on literature system, such as the WOS Core Collection database, is different from a literature review and often applied to analyze the hot spots, research progress and other related issues, as well as to formulate future research directions and strategies. Through visual analysis, the cooperative relationship between organizations, countries and authors and network of research directions, which allows identification of key opinion leaders and future research directions to help in fostering optimised future collaborative networks. Based on the analysis, we recommend that explaining the molecular mechanism and potential role of transcriptional inhibition and immunologic responses to SMARCB1/INI1-negative PDC will be available for preclinical experiments and clinical trials and lead to new therapeutic opportunities for chordoma patients. So the main discussion and attention was on treatment in this bibliometric anslysis, which was similar to the style of several other articles [7,8].

7. Song Y, Zhao F, Ma W, Li G. Hotspots and trends in liver kinase B1 research: A bibliometric analysis. PLoS One. 2021 Nov 4;16(11):e0259240.

8. Waqas A, Salminen J, Jung SG, Almerekhi H, Jansen BJ. Mapping online hate: A scientometric analysis on research trends and hotspots in research on online hate. PLoS One. 2019 Sep 26;14(9):e0222194.

3. From the methodology perspective, authors should emphasize the unique advantages or differences of bibliometric analysis versus systemic review as this analysis is not as common and maybe unfamiliar to most readers. More importantly this could address what unique contribution of this particular article has generated in this field by addressing this issue.

Response: Thank you for your professional suggestion. We have supplemented in the discussion.

4. It is notable that most journals publishing most chordoma paper in the last two decades are relatively low impact. I think it is important for authors to discuss this important finding which may support the future research for rare cancer.

Response: Thank you for your professional suggestion. We have supplemented in the discussion. The level of IF can not directly indicate the value of the research content, because chordoma is a rare disease, not the research hotspot of the whole medical research. Based on the analysis, we recommend that explaining the molecular mechanism and potential treatments will be available for preclinical experiments and clinical trials and lead to new therapeutic opportunities for chordoma patients. According to the journals analysis, IF and JCR partition, Neurosurgery may be the most popular journal in chordoma. So, the journal analysis is just to provide reference for readers to consult relevant literature but not discussed in the manuscript.

5. Although this is well-written manuscript and authors address points clearly and concisely but I strongly recommend authors to use professional English writer (or use Grammars software) to polish up English throughout. One example “ Although chordoma is slow growth, few distant metastases (late metastases), and may delay diagnosis, its local destruction…. “ this sentence does not make sense. Another example : Many places authors use “then” which could be replace by “subsequent” as more professional writing.

Response: Thank you for your professional suggestion. We have conducted the English editing.

Reviewer #2: The authors of this article have attempted to conduct a bibliometric analysis of Chordoma and the current research regarding the same. They also have identified the lacunae in the chinese publications and their lack of of co-operative research. This knowledge is probably relevant to the readers in china and does give some idea to the asian and south asian neurosurgical research communities.

Response: Thank you for your professional comments.

---

## [Decision Letter · Decision Letter 1]

2 Dec 2022

Bibliometric analysis of the global research trends and hotspots in chordoma from 2000 to 2020

PONE-D-21-36906R1

Dear Dr. Dai,

We’re pleased to inform you that your manuscript has been judged scientifically suitable for publication and will be formally accepted for publication once it meets all outstanding technical requirements.

Kind regards,

Farzad Taghizadeh-Hesary

Academic Editor

PLOS ONE

Additional Editor Comments (optional):

Reviewers' comments:

Reviewer's Responses to Questions

**Comments to the Author**

1. If the authors have adequately addressed your comments raised in a previous round of review and you feel that this manuscript is now acceptable for publication, you may indicate that here to bypass the “Comments to the Author” section, enter your conflict of interest statement in the “Confidential to Editor” section, and submit your "Accept" recommendation.

Reviewer #2: All comments have been addressed

Reviewer #3: (No Response)

2. Is the manuscript technically sound, and do the data support the conclusions?

Reviewer #2: Yes

Reviewer #3: Yes

3. Has the statistical analysis been performed appropriately and rigorously? 

Reviewer #2: Yes

Reviewer #3: Yes

4. Have the authors made all data underlying the findings in their manuscript fully available?

Reviewer #2: Yes

Reviewer #3: Yes

5. Is the manuscript presented in an intelligible fashion and written in standard English?

Reviewer #2: Yes

Reviewer #3: Yes

6. Review Comments to the Author

Reviewer #2: The authors have addressed the queries raised by the co-reviwer and probably tried to explain their point of view in the best possible way.

The manuscript has its own merit in the form of explaining recent developments in the diagnostic material based on molecular mechanism and future therapeutic targets.

Reviewer #3: The intended meaning for the sentence "The Chinese institutions and scholars lacked cooperation with their counterparts in other countries." is not clear.

Conclusion needs to be better drafted

7. PLOS authors have the option to publish the peer review history of their article (what does this mean?). If published, this will include your full peer review and any attached files.

Reviewer #2: **Yes: **Rajan Sundaresan Vediappan, Professor, Head & Neck Skull Base Surgery, Department of ENT, unit-1, Christian Medical College, Vellore, TN, India.

Reviewer #3: **Yes: **Amit Agrawal

---

## [Editor Report · Acceptance letter]

5 Dec 2022

PONE-D-21-36906R1 

Bibliometric analysis of the global research trends and hotspots in chordoma from 2000 to 2020 

Dear Dr. Dai:

I'm pleased to inform you that your manuscript has been deemed suitable for publication in PLOS ONE. Congratulations! Your manuscript is now with our production department. 

Kind regards, 

on behalf of

Dr. Farzad Taghizadeh-Hesary 

Academic Editor

PLOS ONE